# Changing Children's Attitudes to Disability through Music: A Learning Intervention by Young Disabled Mentors

Eamonn McCarron [1], Erica Curran [1] and Roy McConkey [2,*]

1 Liberty Consortium, Derry BT48 7RE, Northern Ireland, UK; eamonn@playtrail.com (E.M.); erica@playtrail.com (E.C.)
2 Institute of Nursing and Health Research, Ulster University, Newtownabbey BT37 0QB, Northern Ireland, UK
* Correspondence: r.mcconkey@ulster.ac.uk

**Abstract:** Children with disabilities are at greater risk of social exclusion. In part, this results from the negative perceptions of disability held by their peers. An innovative, school-based project used creative music-making sessions facilitated by young disabled musicians to nurture more positive attitudes among children aged 9 years in four schools, with two classes from each. In all, around 200 pupils were involved in weekly sessions totalling 16 h. Their attitudes to disability were assessed before and after participating in the project, along with the reactions of parents and teachers. Pupils were significantly more disposed to interacting with children with disabilities and to persons with disabilities more generally as well as to having a teacher with a disability. Parents and teachers confirmed the pupils' enthusiasm for the project and the impact it had on them. A core driver for change appeared to be sharing enjoyable musical activities with competent musicians who had disabilities. Further research should explore the potential of mentoring by disabled persons in other arts activities and sports to provide further validation of this approach.

**Keywords:** disability; schools; children; attitudes; intervention; mentoring; music; Ireland

## 1. Introduction

Building a more inclusive society has to start in childhood. This truism is easier said than done, as often even very young children unwittingly absorb the attitudes and prejudices of a previous generation [1]. The stigma associated with disability is still pervasive in many cultures, resulting in children with disabilities being excluded from ordinary activities, which is arguably more true as they grow into adulthood [2]. Moreover, discrimination can be expressed in abusive behaviours, such as name calling and physical violence [3].

In recent years, there has been a growing emphasis on inclusion as evident, for example, in the United Nations Declaration on the Rights of the Child [4]. A particular focus has been on enrolling children with disabilities in mainstream schools [5]. This policy widens their educational opportunities and has been shown to lead to better outcomes socially as well as educationally [6]. However, the success of this policy is dependent on schools making adaptations to their practices and facilities, a lesson that also applies to the promotion of inclusive practices in the wider community.

### 1.1. Literature Review

Thus far, inclusion in schools has been the main focus of research relating to children's attitudes. A recent meta review of 25 literature reviews of inclusive education [5] confirmed that most research interest has been focussed on teacher's attitudes to this form of provision, their professional development, and inclusive education practices. One of the main gaps identified was the lack of attention that had been paid to the voices of students with and without disabilities. However, a systematic review specifically on children's attitudes [2]

identified 37 papers published in the past decade from 2012 to 2019. With few exceptions, prior contact with a person or peer with disabilities was associated with more positive attitudes, whereas no consistent relationships were found with other predictors such as gender and age. In all, 16 intervention studies were identified and nearly all impacted positively on children's attitudes. For example, classroom-based programmes focussed on disability awareness were designed to be delivered as part of the curriculum and generally consisted of taught lessons with accompanying video or simulation activities [7,8]. Others have invited disabled individuals to give presentations in schools, which may include dramas and role-play to actively engage children. Involving disabled peers as mentors for other children has also shown promise [9].

Overall, the majority of interventions used some form of behavioural component to the intervention whereas others focussed on increasing knowledge about disability but with less positive outcomes. By contrast, few researchers included components in their intervention that explicitly targeted the affective dimension of attitudes, such as empathy [2].

### 1.2. Conceptual Framework

The current study brought together these various intervention approaches into a novel project named 'Project Sparks', which is aimed at nurturing more positive attitudes with 9- and 10-year-olds. The intervention was built around personal interactions between the pupils and young people with a disability as this has proven to be an effective means for creating more positive attitudes across different marginalised groups with children and adults alike [10]. Moreover, the three components of attitudes—cognition, behaviour, and affect—would be addressed with a particular focus on the latter [11]. Music was chosen as the medium for changing attitudes as it has been shown to improve social and engagement outcomes [12], nurture positive emotions, and reduce psychological stress [13].

To this end, the project trained young disabled adults to mentor primary school children. It was thought that mentorship within the context of participatory music would foster an enjoyable, interactive learning environment, in which shared experiences can be developed non-verbally if necessary [14]. Furthermore, the mentors' competency in music performance would counter negative stereotypes of disability often held by children [15].

### 1.3. The Study

This paper focuses on the evaluations undertaken to identify the impact of this novel approach. This involved assessing the children's perceptions of disability before and after taking part in the project, the feedback from their parents, and the perceptions of the children's teachers. This project was repeated in four schools with two classes in each and involved nearly 200 pupils.

In summary, the aims of the study were:

1. To determine the impact on pupil's attitudes to disability, following 16 hours of music workshops co-facilitated by young disabled mentors.
2. To obtain teachers' and parents' views on the effectiveness of the intervention.

## 2. Materials and Methods

### 2.1. Project Sparks

The project was created by two freelance music teachers as a means of promoting the artistic and musical talents of young people with disabilities and to challenge the expectations society had of their capabilities. Over a series of weekly workshops, the musical and performance skills of the young adults with disabilities were honed through mentoring and peer support. The main forms of training activities were role-play, video-analysis of other mentors teaching, and practice workshops, in which the mentors trialled their skills with small groups of children from local primary schools. The mentors were trained in promoting children's creativity by learning the foundational concepts of music through singing, dance, and a range of percussion instruments.

From an initial cohort of 22, 10 trainee mentors dropped out, mainly due to the difficulty of the training and reluctance to perform and speak in front of others. The remaining 12 not only increased their competence as music mentors but they developed a common 'ownership' of their identity as disabled individuals; openly speaking about their disability, perceiving it as a largely positive feature in their lives [15].

Funding was then obtained from a charitable foundation to undertake music sessions for children attending four schools in a city with high levels of social deprivation. Within the schools enlisted, the percentage of pupils entitled to free school meals (a proxy for social deprivation) ranged from 41% to 85% and had few pupils learning musical instruments (0–6%). Whilst all four schools were predominantly attended by non-disabled children, a proportion of pupils had special educational needs, although the principals reported limited opportunities for social integration, and name-calling toward disabled pupils was expressed as an issue in one school.

The music sessions provided by Project Sparks had the dual aims of increasing the children's participation and confidence in music making, but also to nurture more inclusive beliefs toward disability and social diversity in general.

The 12 disabled musicians became mentors alongside the co-creators of the project (the first and second authors). Eight classes took part, involving around 200 pupils. Each class received 16 face-to-face hours of music sessions although a portion of the classes were modified due to COVID-19 restrictions. Some were held in their school, others in a performance space available to the project, and for one class, the sessions were delivered remotely via the Internet.

The lessons developed pupils' basic understanding of music through a range of guided, discovery-based activities that cumulatively introduced them to each 'building block' of music (rhythm, tempo, melody, and timbre) using xylophones, djembes, ukuleles, physical movement, and voice. Pupils' exploration of each building block developed their creative, aural, and motor skills as they composed and appraised one another's creative ideas.

The activities were adapted for each class to inculcate the key academic skills deemed most pertinent by the teacher of each particular class. For example, in response to one class who expressed difficulty understanding fractions, the project staff devised simple rhythmic activities that served as an alternative instructional pathway. These activities contextualised fractions by making links to musical note values. Furthermore, bespoke songs were composed to help classes learn key literacy concepts such as homophones, adjectives, verbs, and nouns. The children were then facilitated in creating unique melodies and rhythms to fit around each song.

Pupils who were 'failure-averse' were invited to embrace a growth mindset by extracting learning from their mistakes and celebrating the process of creativity equally as much as the end-products.

The mentors' role involved:

- Outlining the learning outcomes associated with each classroom activity;
- Role-modelling success in music composition and performance;
- Facilitating group work with pupils to advance their experimentations with the musical building blocks and foster sharing of ideas;
- Providing pupils with praise and constructive feedback to improve their compositions;
- Answering questions about their disability during informal chat sessions.

Mentors were also involved in the planning for and reflection on classes through training, in which the project leaders scaffolded their development by analysing video-footage of their mentoring from recent classes and role-playing the activities designed for the school pupils.

### 2.2. Participants

Disabled Mentors: Of the 12 disabled mentors, 7 were female and 5 male, with ages ranging from 19 to 28 years. They had a variety of impairments with some co-occurring: six were on the autism spectrum and six had learning disabilities, two had cerebral palsy,

one spinal bifida, and one Down syndrome. Six had attended special schools and six mainstream schools. Their chosen musical specialisms were singing (*n* = 6); drumming (*n* = 5); dance (*n* = 3); and song writing (*n* = 1).

Children: The 171 children (51% male) who took part in the evaluation were in fourth grade of primary schools aged 9 to 10 years. (This represented around 85% of those in the classes with others missing due to absences when information was gathered). The pupils were asked to report their prior experience of disability and 38% had a person in their family with a disability; 90% knew there were children with a disability in their school; and 38% had such a child in their class. Moreover, 30% reported that one of their close friends had a disability. Outside of school, 16% reported they had daily contact with someone with a disability; 10% had weekly contact; but 30% only sometimes had contact and 43% had no contact.

*2.3. Evaluation*

Three groups of informants were used to assess the impact of the project on children's attitudes.

Children: The children's attitudes to disability were assessed before and after participating in Project Sparks. They completed a computer-based questionnaire that consisted of a series of items that had been used in a previous study [16] (see Table 1 below). This was undertaken individually in their classrooms with assistance from their teacher or classroom assistant if required. Children chose one response from a five-point Likert scale consisting of Big No, No, Maybe, Yes, and Big Yes.

Prior to answering the questions, the children were told: "*I want you to think about a person around the same age as you, who has a disability. There are different types of disability. Sometimes people can be physically disabled which means they have a part of their body which does not work properly. So maybe their legs do not work and they cannot walk so they have a wheelchair or use sticks. Other people may not be able to see or hear, or they can have a learning disability. This means some children find it hard to learn things and they find it more difficult than other children. People with learning disabilities sometimes behave differently too. Many people with disabilities have been like that since they were born. They might have to get extra help.*"

The children completed the items in relation to a child with a disability. They were then asked to rate the same items in relation to an African child who had joined their class. This would provide a contrast to their responses to a child with disability.

Principal components analyses on the children's ratings identified that the items mainly loaded on one factor which accounted for 42% of the variance on ratings to a child with a disability and also to ratings for an African child. The Cronbach alphas for summated ratings of children with disabilities was 0.808 and for African children was 0.820, which are indicative of reasonable internal reliability on the items.

Parents: The children's parents were sent a text message about the project by the class teacher and given a link to an online questionnaire to obtain their reactions to the project. The parents also rated a series of statements with four options provided. Sample items were: How enthusiastic was your child about the project? (Not at all; Did not seem bothered; Somewhat enthusiastic; Very enthusiastic); Most of the mentors on the project had a disability—did your child talk about this? (No; A little; Sometimes; A lot); How important is it for children to experience being taught by people with a disability? (Not important; Somewhat important; Important; Very important).

Teachers: Telephone interviews were conducted with the class teachers and two of the headteachers of the participating schools. These were undertaken—before and after classes participated in the project—by the third author who had no involvement with its presentation. Written transcripts were made of the interviews conducted after the project's completion (totalling 91 min). Codes were first assigned to the teachers' comments by the third author, and these were then grouped by themes that were revised to ensure that all relevant comments had been included [17]. These were then validated by the project leaders (the first and second authors).

## 3. Results

### 3.1. Pupils' Perceptions of Disability

Table 1 presents the percentage of children who responded Yes to the items listed before and after participating in the project (note the items relating to being afraid were reverse coded to reflect No responses). These ratings were repeated for an African child although the order of the items was ranging from 1 to 5 with Yes scoring 4 and Big Yes 5. (for 'afraid' items, reverse scoring was used with different for these ratings).

**Table 1.** The percentage of children responding Yes to the items relating to 'disabled child' and 'African child' before and after taking part in Project Sparks.

| Items (The Words 'African' or 'Disabled' were Inserted) | Disabled Child Before | Disabled Child After | African Child Before | African Child After |
|---|---|---|---|---|
| I would enjoy having a ___child in my class. | 64.7% | 87.0% | 80.6% | 86.2% |
| I would be pleased if a ____child invited me to play with him or her at school. | 77.0% | 87.0% | 80.0% | 86.2% |
| I would feel good doing a school project with a ____child. | 72.4% | 85.3% | 85.3% | 88.7% |
| I would try to stop people in my school making fun of a ____child. | 97.6% | 95.1% | 95.9% | 95.9% |
| (African) Children (with disabilities) can do just as well at school as (Irish) children (with no disabilities). | 81.2% | 88.7% | 84.8% | 93.5% |
| If a ____ child was feeling left out, I would ask them to join in. | 94.5% | 96.8% | 94.1% | 96.7% |
| I would invite a ____ child to come to my house to play. | 60.6% | 67.4% | 63.6% | 65.9% |
| I would know what to say to (African) children (with disabilities). | 43.5% | 59.4% | 69.4% | 73.2% |
| I would feel afraid of having a (African) child (with disabilities) in my class. | 82.3% | 89.4% | 87.1% | 84.6% |

The children's ratings before taking part in the project for children with disabilities and African children were similar on items relating to making fun of and being left out. This may reflect the anti-bullying work undertaken by schools. For the other items relating to more personal contact, the pupils were more positively disposed to interacting with African children. This was most marked in the item: "I would know what to say".

After the project, the children were more inclined to answer Yes, especially in relation to children with disabilities, but similar effects were also present for African children.

This was further confirmed when a summary score was calculated on children's rating across the 9 items (No scoring 4 and Big No scoring 5). A maximum score of 45 was indicative of more positive attitudes. Table 2 summarises the mean scores summated across the pre and post rating given to items relating to 'disabled child' and 'African child'.

**Table 2.** The means and (standard deviations) of summated ratings given to 'disabled child' and to 'African child' before and after taking part in Project Sparks.

| Disabled Child Before | Disabled Child After | African Child Before | African Child After |
|---|---|---|---|
| 37.03 (4.83) | 38.83 (4.57) | 38.42 (4.81) | 39.07 (4.65) |

The mean scores at pre-test given to children with disabilities was significantly lower than those to African children (t = 4.774, df 165, $p < 0.001$: Cohen's d = 0.383) although the effect size as indicated by Cohen's d is considered to be small.

However, after taking part in the project, the ratings for the two sets of children were not significantly different, with ratings to children with disabilities increasing more than those to African children. (It should be noted that ceiling effects may be operating as the mean scores are close to the maximum score of 45).

It is worth noting that the correlations between ratings given to the two groups was not particularly high although statistically significant (before r = 0.699; after r = 0.612). This would suggest that children were perceiving differences in their interactions with the two groups.

As a further test of children's attitudes, they were asked to rate five items relating to contacts outside of school and largely with adults with disabilities as shown in Table 3.

**Table 3.** The percentage of ratings of 'Yes' and 'Big Yes' before and after the project.

| Item | % Before | % After |
|---|---|---|
| I would be pleased if a person with disabilities lived next door to me. | 65.3% | 85.3% |
| I would be happy to sit beside a disabled person on the bus or train. | 55.3% | 78.1% |
| I would go to a hairdresser or barber who was disabled. | 41.8% | 58.5% |
| I would help a disabled person to cross the road. | 97.1% | 95.9% |
| I would be afraid to talk to a person with a disability. | 74.7% | 80.5% |

On all the items, an increase in Yes ratings occurred after the project.

*3.2. Perceptions of Teacher with Disabilities*

The children were asked to rate a single item: "It would be good to have a teacher with disabilities in my school". The same rating scale was used as for the attitude items. Before the project, 43.6% chose Yes or Big Yes (with 45.3% selecting Maybe). After the project, the equivalent percentages were 62.6% Yes (31.7% Maybe).

A subset of pupils (*n* = 116) who took part in the project later were asked some additional questions to explore this issue further. Table 4 summarises their responses to items asked (note: the wording was changed in the questions asked after the project to the "leaders in the project"). For the first item, the options provided were "Not good; OK; Good; Very good; Excellent". The last three items were scored out of 10.

**Table 4.** The means and (SDs) given to people/leaders with disabilities.

| Items | Before | After |
|---|---|---|
| What kind of teacher would you imagine a disabled person being? % rated excellent | 22.8% | 52.0% |
| How smart or stupid do you imagine disabled people to be? | 7.65 (1.95) | 9.16 (1.59) |
| How musically talented or not talented do you think disabled people are? | 7.87 (2.13) | 9.74 (0.66) |
| How much would you want a disabled person to teach you? | 6.48 (2.57) | 9.40 (1.33) |

As the table shows, there were large increases in all the ratings given to the mentors with ceiling effects present as the scores after participating in the project came close to the maximum of 10.

*3.3. Parents' Responses*

Responses were received from 54 parents. In all, 80% of parents reported that their child had spoken to them about the project 'a great deal' and 89% mentioned how their child was very enthusiastic about it. Furthermore, 85% of the children had talked to their parents about being taught by people with a disability and 80% of parent felt it was 'very

important' for children to have this experience with the other 20% rating it 'important'. In all, 95% stated they would definitely recommend the project to other parents and schools.

Nearly half the parents added a comment generally praising the project. One commented "*This project made my child very excited and stimulated going to school*". Another said "*She loved the leaders (mentors) and how they taught songs and instruments. To me as a parent it is an excellent way of teaching children equality and respect and empathy for each other.*"

*3.4. Teachers' Responses*

The teachers were also enthusiastic about the project and the contribution that the mentors had made to the lessons. Four themes were apparent in a thematic content analysis of their responses. First, they spoke of the relationships they had developed with the children. "*At the beginning, they (children) were quite shy and quite afraid to actually approach any of the leaders and whenever leaders would come over to chat, they were still quite shy. By the end, there was none of that at all. They were just like going over –to ask any questions that they wanted to ask. They were like 'How was your childhood, with a disability?' 'I feel that they really benefitted'.*" (CH).

A second theme was the mentors' competence as music instructors. Another teacher recounted how when told that the mentors had a disability, a pupil responded: "How are they going to do anything with us", but he remarked on the change: "*I thought the awareness that the children got of working with people with different learning needs was unbelievable. It changed their whole perception of what they thought that these people could do.*" (PG).

A third theme was the insight it had given the children into disability: "*gaining an understanding of why and how we are different, it is nothing to be afraid of*" (FK). Another teacher recalled how the leaders encouraged the children to ask them questions: "*they told (the children), no matter what, if you think your question is silly, or if it's going to be hurtful to us, just you ask it. To give them that idea of what it was like to have maybe the likes autism or to have a physical disability*" (PG). It was not only the children that got new insights. A teacher admitted: "*I also learned about disabilities too. There were things I didn't know so, it helped me as well. As a teacher I got loads out of it and I would really recommend it to other teachers.*" (FP).

The mentors seemed to act as role models for children with special needs in the class. A head teacher explained: "*Our kids with specific learning needs, or with say autism for example, they're very reluctant to get involved in certain things in class because of their learning needs or their difficulties. But in the project they felt important and valued. And they realised that, 'my goodness look at these guys teaching us how to do this, this could be me in the future doing this'. Both the confidence and the ability to excel in life, regardless of your background or your educational or your social position, that was instilled throughout the programme*".

Finally, all the teachers commented on how much enjoyment the children got out of the project. "*As soon as you went into the room, you couldn't help but smile. There was such a real positive energy about the place all the time*" (FP).

## 4. Discussion

This project, which was aimed at nurturing positive attitudes to people with disabilities, was unique in two main ways. It enlisted disabled young people as music mentors in a school-based programme that brought the pupils into personal and active contact with individuals with various disabilities over a period of weeks. This stands in contrast with previous approaches that are usually led by teachers or invited guests, albeit with a disability, who make a one-off visit to the classroom and have little interaction with the children [8].

The second feature was the use of music as a context for their interactions. In one sense, this may have placed the pupils at somewhat of a disadvantage due to their limited experience of music-making and creating their own songs. Yet, it provided them with a challenge but in a relaxed and informal context when they engaged in shared activities with one another and with supportive mentors who carefully built their confidence to experiment and to enjoy making music together.

A few other aspects are worth bearing in mind. The project was relatively short which should make it manageable to include within the school curriculum. The impact of the project was replicated in four different schools and with two classes in each. However, the classes were drawn from the same age group and modification may be required to use it with younger or older pupils. The schools serve families living in areas of high social deprivation which is sometimes linked to more negative attitudes toward disability [18]. This could be considered a more stringent test of the methodology than had it been conducted in schools located in more affluent areas.

The findings confirm the important role that personal contact plays in changing perceptions of disability, even for young children [10]. However, our experiences suggest that the contact needs to be enjoyable and meaningful for the children and one that demonstrates the competence of the disabled persons. The relationships the mentors created with the pupils was evident from their responses and those provided by the parents and teachers. Indeed, the formation of relationships may give some insight into why personal contact is a driver for change across the three domains of attitudes but especially in creating positive affect to persons with disability [11]. Moreover, the disabled mentors also spoke of the confidence boost they gained from participating in the project and the increased status they gained as music mentors. Their enthusiasm and active participation also contributed to the building of relationships with the children.

The study has limitations. A specific age group was targeted, and the intervention may need to be adjusted for use with teenagers or preschool children. The adoption of the project by schools also needs to be tested. Furthermore, the contribution of music per se to attitude change remains speculative as a comparison is not available with other activities that the mentors might have undertaken instead of music, such as talks or role plays which previous studies have done [8].

Nonetheless, there were indications to suggest that music played an important role. The children were active participants throughout, they engaged in shared music-making with their peers and the mentors, and they experienced success as they performed their songs for their class. The social bonding that music created is arguably the more important benefit rather than music per se [19], and indeed this might be attainable through other art forms such as drama [20] or in sports [21].

Future research could usefully elucidate the variations in social bonding that arise from different shared activities and for this to happen with different age groups. Such strategies would increase the opportunities for children with disabilities to become more included within schools and other community activities, thereby realising the aspiration contained in the UN Declaration on the Rights of the Child (Article 23) [4]: "(they) should enjoy a full and decent life, in conditions which ensure dignity, promote self-reliance, and facilitate the child's active participation in the community".

## 5. Conclusions

Children's attitudes to disability were positively impacted by participation in a music-making project facilitated by young adults with a range of disabilities. The children's self-reports were confirmed by parental accounts and those of their teachers. The project was replicated in four schools and with two classes in each school. Core drivers for change were the sharing of enjoyable musical activities with competent musicians who had disabilities.

**Author Contributions:** Conceptualisation, E.M. and E.C.; methodology, E.M., E.C. and R.M.; formal analysis, R.M.; investigation, R.M.; resources, E.M. and E.C.; data curation, E.M.; writing—original draft preparation, R.M.; writing—review and editing, R.M., E.M. and E.C.; project administration, E.M.; funding acquisition, E.M. and E.C. All authors have read and agreed to the published version of the manuscript.

**Funding:** Project Sparks received funding from the Paul Hamlyn Foundation to undertake the work in schools and to cover the costs of evaluating its impact.

**Institutional Review Board Statement:** Ethical review and approval were waived for this study, as under UK regulations, it was classed as an evaluation/audit of service delivery.

**Informed Consent Statement:** Informed consent was obtained from all participants involved in the study and assurances were given about the confidentiality of the information they provided.

**Data Availability Statement:** The data reported in this study are available on reasonable request from the corresponding author.

**Acknowledgments:** Our thanks to the schools and the head teachers for facilitating the collection of information from the pupils and parents.

**Conflicts of Interest:** The authors declare no conflict of interest.

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
