# Peer review of "Changing Children’s Attitudes to Disability through Music: A Learning Intervention by Young Disabled Mentors"

_disabilities, doi:10.3390/disabilities2010008_

Round 1

Reviewer 1 Report

This is an interesting paper, with an important topic on social acceptance of disabled children, describing an intervention, helping to overcome the negative perceptions of disability. The paper is well written, but it is not well grounded into the extant literature. Authors should consult similar research and report how their research fits into the existing literature, and what it adds to the existing body of knowledge.

In terms of specific recommendations, I would like to emphasize that the authors reference several well-chosen systematic reviews and meta analyses in their introduction. However, they did not seem to have thoroughly read through them, but rather 'casually' mention those studies from the field of inclusive education, without summarizing and reporting the previous research on the specific topic of disabled persons' motivation and motivation-related interventions for inclusive education. I would suggest that this discussion is conducted in an additional section - to be called 2. Literature overview, and placed between 1. Introduction and 3. Materials and methods. For a scientific article, it is completely unacceptable to skip the literature overview section and briefly present some references in the Introduction section. Introduction should report on the research problem, why it is important and interesting for the readers, as well as provide some initial description of the entire research project. In the discussion, please include at least several sentences on how your results contribute to the body of knowledge on how music and art motivate disabled children to participate in the inclusive education - try to place the benefits of your project in the findings of previous research.

These are some basic requirements for any scientific study, which might have led the reviewer toward the 'major revision' or even 'reject' recommendations. However, I believe that your topic is extremely important for the practice of inclusive education, where such innovative projects are not common. As not to discourage you from working on improvement of this paper, the recommendation is, thus, 'minor revision', but I do urge you to put more effort into the literature review and discussion sections, which will make your paper publishable.

Author Response

This is an interesting paper, with an important topic on social acceptance of disabled children, describing an intervention, helping to overcome the negative perceptions of disability. The paper is well written, but it is not well grounded into the extant literature. Authors should consult similar research and report how their research fits into the existing literature, and what it adds to the existing body of knowledge.

Many thanks for your affirmation about the study and your helpful suggestion.

In terms of specific recommendations, I would like to emphasize that the authors reference several well-chosen systematic reviews and meta analyses in their introduction. However, they did not seem to have thoroughly read through them, but rather 'casually' mention those studies from the field of inclusive education, without summarizing and reporting the previous research on the specific topic of disabled persons' motivation and motivation-related interventions for inclusive education. I would suggest that this discussion is conducted in an additional section - to be called 2. Literature overview, and placed between 1. Introduction and 3. Materials and methods. For a scientific article, it is completely unacceptable to skip the literature overview section and briefly present some references in the Introduction section. Introduction should report on the research problem, why it is important and interesting for the readers, as well as provide some initial description of the entire research project.

In the discussion, please include at least several sentences on how your results contribute to the body of knowledge on how music and art motivate disabled children to participate in the inclusive education - try to place the benefits of your project in the findings of previous research.

These are some basic requirements for any scientific study, which might have led the reviewer toward the 'major revision' or even 'reject' recommendations. However, I believe that your topic is extremely important for the practice of inclusive education, where such innovative projects are not common. As not to discourage you from working on improvement of this paper, the recommendation is, thus, 'minor revision', but I do urge you to put more effort into the literature review and discussion sections, which will make your paper publishable.

Many thanks for your helpful comments which we have addressed in the light of your comments.  Our focus primarily was on the literature on attitude change in children rather than on inclusive education per se.  However we have included more details from the extensive literature reviews we have cited and the need identified in them to focus on children's perceptions that have been given little attention in the past. 

The main theoretical framework informing our study was the role of personal contact (or lack of it) in attitude formation and change.  Our findings confirm and extend the role of contact as we expand in the discussion. 

Our priority was to conduct an evaluation of a practical classroom-based intervention: a feasibility study primarily.  We are conscious of its limitations and have noted suggestions for further research. 

Reviewer 2 Report

The article has a good insight into ways in which society can start addressing the disparities of students with disabilities via music. The major issue with the paper is that the authors, who are working to reduce the negative attitudes and interactions with people with disabilities, do not use person-first language throughout the document. The tables also need to have different headings. They should stand alone yet the headings (if they were truly stand-alone) sound like the authors are looking at people before they had a disability and after they became disabled.  The same is true for the label of Africa.  Finally, the authors should have a discussion regarding limitations of the study.

Author Response

The article has a good insight into ways in which society can start addressing the disparities of students with disabilities via music. The major issue with the paper is that the authors, who are working to reduce the negative attitudes and interactions with people with disabilities, do not use person-first language throughout the document.

The phrase 'disabled child' came from a previous questionnaire.  It was also a term chosen by the tutors as it more commonly used in Irish society.  The reviewer will be aware that it is a term reflective of the social model of disability.  

The tables also need to have different headings. They should stand alone yet the headings (if they were truly stand-alone) sound like the authors are looking at people before they had a disability and after they became disabled.  The same is true for the label of Africa. 

Thanks for pointing this out and we have clarified the titles accordingly. 

Finally, the authors should have a discussion regarding limitations of the study.

The limitations have been noted.

Reviewer 3 Report

This is short paper of moderate importance. The methodology seems to be sound, but some aspects could be explained more in detail. The weakest part of the paper is the rather limited reference list and the very limited theoretical and empirical framework. The paper could benefit from a broadening of this framework. I do not know whether there are word count restrictions for this paper, but a more substantial elaboration of the aims and used methodology should be welcome. I would suggest to accept the paper for publication, given that the comments below are addressed appropriately.

General remarks

  • The language use is OK and the style of writing if fluent.
  • The structure of the paper is quite coherent.
  • The number of participants is quite substantial which gives the paper some generalizing power.
  • The methodology seems to be sound, but the assessment method (scores) should be explained more in detail.
  • The thematic content analysis should also be explained more clearly.
  • The theoretical and empirical background is quite limited.
    Some claims are rather hypothetical and tentative.

Detailed comments

  • page 1: keywords: some of them are rather trivial (children, music)
  • page 4: please elaborate somewhat more in detail about the principal component analysis. Which components were found? Describe the methodology somewhat more in detail.
    page 4: parents: what are the four options? This is not clear.
  • page 5, table 1 caption: “children” instead of “children”
  • page 5, table 2: please explain the ratings. How to interpret the scores? Which units are used?
  • page 6: 3.2: explain the rating scale more in detail. Which were the possible ratings?
  • page 6: 3.4: please provide more details about the thematic content analysis. Explain the methodology somewhat more.

Author Response

This is short paper of moderate importance. The methodology seems to be sound, but some aspects could be explained more in detail. The weakest part of the paper is the rather limited reference list and the very limited theoretical and empirical framework. The paper could benefit from a broadening of this framework. I do not know whether there are word count restrictions for this paper, but a more substantial elaboration of the aims and used methodology should be welcome. I would suggest to accept the paper for publication, given that the comments below are addressed appropriately.

We appreciate your helpful comments and although we did not want the paper to be overlong, we welcome the opportunity to provide additional information as noted below.

General remarks

  • The language use is OK and the style of writing if fluent.
  • The structure of the paper is quite coherent.
  • The number of participants is quite substantial which gives the paper some generalizing power.
  • The methodology seems to be sound, but the assessment method (scores) should be explained more in detail.

We have included further details

  • The thematic content analysis should also be explained more clearly.

This has been done and a reference to the methodology used has been provided. 

  • The theoretical and empirical background is quite limited.
    Some claims are rather hypothetical and tentative.

Extra details are provided as per other reviewer's comments

Detailed comments

  • page 1: keywords: some of them are rather trivial (children, music)

The keywords have been revised

  • page 4: please elaborate somewhat more in detail about the principal component analysis. Which components were found? Describe the methodology somewhat more in detail.

More details have been given

  • page 4: parents: what are the four options? This is not clear.

The four options are described.

  • page 5, table 1 caption: “children” instead of “children” Corrected
  • page 5, table 2: please explain the ratings. How to interpret the scores? Which units are used?

This information is provided

  • page 6: 3.2: explain the rating scale more in detail. Which were the possible ratings?

This information is provided

  • page 6: 3.4: please provide more details about the thematic content analysis. Explain the methodology somewhat more.

This information is provided